# Li_7_La_3_Zr_2_O_12_-co-LiNbO_3_ Surface Modification Improves the Interface Stability between Cathode and Sulfide Solid-State Electrolyte in All-Solid-State Batteries

**DOI:** 10.3390/membranes13020216

**Published:** 2023-02-09

**Authors:** Shishuo Liang, Dong Yang, Jianhua Hu, Shusen Kang, Xue Zhang, Yanchen Fan

**Affiliations:** 1State Key Laboratory of Molecular Engineering of Polymer, Department of Macromolecular Science, Fudan University, Shanghai 200438, China; 2Sunwoda Electric Vehicle Battery Company, Shenzhen 518107, China; 3College of Resources and Environment, Jilin Agricultural University, Changchun 130118, China; 4Petro China Shenzhen Renewable Energy Research Institute Co., Ltd., Shenzhen 518000, China; 5CNPC Shenzhen New Energy Research Institute Co., Ltd., Shenzhen 518000, China; 6SUSTech Academy for Advanced Interdisciplinary Studies, Department of Materials Science & Engineering, Southern University of Science and Technology, Shenzhen 518055, China

**Keywords:** solid-state batteries, LiCoO_2_, sulfide solid-state electrolyte, LiNbO_3_-coating, LLZO-coating

## Abstract

With the rapid development of energy storage and electric vehicles, thiophosphate-based all-solid-state batteries (ASSBs) are considered the most promising power source. In order to commercialize ASSBs, the interfacial problem between high-voltage cathode active materials and thiophosphate-based solid-state electrolytes needs to be solved in a simple, effective way. Surface coatings are considered the most promising approach to solving the interfacial problem because surface coatings could prevent direct physical contact between cathode active materials and thiophosphate-based solid-state electrolytes. In this work, Li_7_La_3_Zr_2_O_12_ (LLZO) and LiNbO_3_ (LNO) coatings for LiCoO_2_ (LCO) were fabricated by in-situ interfacial growth of two high-Li^+^ conductive oxide electrolytes on the LCO surface and tested for thiophosphate-based ASSBs. The coatings were obtained from a two-step traditional sol–gel coatings process, the inner coatings were LNO, and the surface coatings were LLZO. Electrochemical evaluations confirmed that the two-layer coatings are beneficial for ASSBs. ASSBs containing LLZO-co-LNO coatings LiCoO_2_ (LLZO&LNO@LCO) significantly improved long-term cycling performance and discharge capacity compared with those assembled from uncoated LCO. LLZO&LNO@LCO||Li_6_PS_5_Cl (LPSC)||Li-In delivered discharge capacities of 138.8 mAh/g, 101.8 mAh/g, 60.2 mAh/g, and 40.2 mAh/g at 0.05 C, 0.1 C, 0.2 C, and 0.5 C under room temperature, respectively, and better capacity retentions of 98% after 300 cycles at 0.05 C. The results highlighted promising low-cost and scalable cathode material coatings for ASSBs.

## 1. Introduction

Lithium-ion batteries (LIBs) applied to 3C, electric vehicles, and other fields [1] constitute potential safety hazards and have limited energy density because of the use of liquid organic electrolytes and graphite cathodes [2], which hinder the development of LIBs. Moreover, with the fast growth of the demand for electric vehicles, the safety and energy density of LIBs have attracted more and more attention. All-solid-state batteries with solid-state electrolytes and lithium metal anodes have a preponderance of safety and energy density compared with their conventional liquid counterparts. All solid-state batteries are considered the most promising next-generation lithium-ion batteries and attract the attention of various companies and research teams [3,4]. However, all solid-state batteries cannot be applied widely because of various bottlenecks, such as low Li^+^ ion conductivity, interface side reaction, and space-charge-layer [5].

Solid-state electrolytes are indispensable to realizing all-solid-state batteries. Numerous effects have been devoted to developing solid-state electrolytes, such as oxide solid-state electrolytes, sulfide solid-state electrolytes, and solid polymer electrolytes. Oxide solid-state electrolytes include perovskite-type LLZO, LISICON-type, NASICON-type, and garnet-type LLZO [6]. The Li^+^ conductivity of oxide solid-state electrolytes is higher than that of solid polymer electrolytes and lower than that of sulfide solid-state electrolytes. The electrochemical window of oxide solid-state electrolytes is the widest in all solid-state electrolytes. However, oxide solid-state electrolytes are brittle and have bad interface contact. Solid polymer electrolytes without the problems of liquid electrolyte leakage and flammability have the advantages of decent interface contact, good electrochemical stability, easy fabrication, and economic availability. The low Li^+^ conductivity of solid polymer electrolytes hinders their application [7]. Sulfide solid-state electrolytes with the highest Li^+^ conductivity and narrowest electrochemical window include LPSC, Li_10_GeP_2_S_12_ (LGPS), and so on [8]. Sulfide solid-state electrolytes are a hot research topic in solid-state batteries, because their high Li^+^ conductivity is similar to those of liquid electrolytes. Sulfide solid-state electrolytes have critical challenges that need to be solved for practical battery applications, such as poor contact among the solid particles, lithium dendrite growth in SEs, and volume change of cathode materials. Interface problems of sulfide solid-state electrolytes with lithium anodes and cathodes are challenging. Batteries will deteriorate rapidly due to side reactions between sulfide solid-state electrolytes and lithium anode, and the space charge layer between sulfide solid-state electrolytes and cathode [9]. All these issues must be solved in the future.

Cathode materials are one of the primary factors restricting the specific capacity and energy density of LIB. However, the detrimental interfacial reactions, poor solid–solid contact, and volume change impede the development of all-solid-state batteries [9]. LCO, which was discovered in 1980 and commercialized by SONY in 1991, is one of the main cathode materials. Despite being theoretically outstanding (274 mAh/g), the stable delithiation of LCO is limited to about 0.5 Li per formula at around 4.2 V (vs. Li^+^/Li), yielding a practical capacity of 140 mAh/g. The lattice structure of LCO with high discharging potential [10] is of a stably layered α-NaFeO_2_ type with a *R*$\overset{-}{3}$*m* space group [11]. The oxygen atoms in the crystal lattice of LCO are arranged with a cubic close-packed framework, forming an edge-shared octahedron. Li^+^ and Co^3+^ are alternately located in the positions of 3a and 3b of cubic close-packed octahedrons, respectively, while O^2−^ is located in the position of 6c, forming a layer structure with alternate layers of alkali metal and transition metal. Due to the specific crystal structure, LCO possesses the merits of having a high discharge platform, remarkable specific capacity, and high tap density. Extensive studies performed on LCO at high charging voltages revealed two main degradation mechanisms. The first one is related to the onset of significant phase transitions occurring in the cathode bulk when the electrodes are charged beyond 4.5 V. These phase transformations are followed by changes in the lattice dimensions, which induce additional particle strain and the formation of cracks. The second degradation mechanism relates directly to the surface, originating from electrode–electrolyte interactions. Charging to a high voltage results in the dissolution of Co and O_2_ release, and leads to the decomposition of the electrolyte, resulting in the development of passivation layers that increase the electrodes’ impedance.

Due to the narrow electrochemical window, sulfide-based all-solid-state batteries suffer from poor solid–solid contact and serious interfacial reactions between LCO and sulfide electrolytes. The side reaction between cathode materials and sulfide-based electrolytes passivate the cathode/electrolyte interface. The electrochemical volume changes in cathode materials cause microstructure changes and even contact loss. Passivation and contact loss of the cathode/electrolyte interface induce growth in interfacial resistance and capacity loss. Surface coating of cathode materials is a simple and effective way to solve this problem. To enhance the Li^+^ transfer kinetics, various in situ coating strategies for the surface modification of cathode materials were developed by solution-based methods. Furthermore, the fabrication of thin coating layers is also beneficial for shorter Li^+^ transfer. As reported in many previous studies, the stable surface coating could suppress the interfacial side reactions and broaden the electrochemical window. Up to now, the ion conductors are usually used as the surface-coating materials, such as LiNbO_3_ [12], Li_2_SiO_3_ [13], Li_2_ZrO_3_ [14], Li_3_PO_4_ [15], Li_4_Ti_5_O_12_ [16], LiTaO_3_ [17], Li_3_B_11_O_18_ [18], Li_3-x_B_1-x_C_x_O_3_ [19], LiNb_0.5_Ta_0.5_O_3_ [20], Li_0.35_La_0.5_Sr_0.05_TiO_3_ [21], and Li_2_CoTi_3_O_8_ [22]. However, the low ionic conductivity (10^−6^–10^−9^ S/cm) of these surface-coating materials significantly restricts the electrochemical performance of all-solid-state batteries [23]. Moreover, surface-coating procedures are very complex. The distribution of surface-coating materials is usually uniform. To achieve solid-state batteries with energy densities above 300 Wh/kg, the content of solid-state electrolytes in the cathode needs to be below 15%, but now most solid-state batteries have 30% solid-state electrolyte in the cathode, which cannot achieve high energy density solid-state batteries [20].

Herein, we fabricated the Li_7_La_3_Zr_2_O_12_-co-LiNbO_3_ coating LCO using wet-chemical approaches to improve the interfacial stability between the cathode and sulfide electrolyte. Li_7_La_3_Zr_2_O_12_ and LiNbO_3_ are fast ion conductors and have high ion conductivity. LiNbO_3_ are inner-coating materials, which could improve the rate capacity [24]. Li_7_La_3_Zr_2_O_12_, with high chemical and electrochemical stability, is the outer-coating material, which benefits the interfacial stability. Two surface coating layers could better suppress interfacial side reactions. Wet-chemical approaches can easily make an even surface coating. Structural characterization reveals that the coating process does not destroy the LCO lattices and instead forms a uniform LLZO and LNO coating layer. LLZO&LNO@LCO has more homogeneous particle size distributions and a higher specific surface area than LCO. Additionally, the volume resistance of LLZO&LNO@LCO is lower than LCO because of the Li^+^ conductivity of LLZO and LNO coatings. ASSBs containing LLZO&LNO@LCO have a significant advantage in long-term cycling performance and discharge capacity compared with those assembled from uncoated LCO. LLZO&LNO@LCO||LPSC||Li-In delivers discharge capacities of 138.8 mAh/g, 101.8 mAh/g, 60.2 mAh/g, and 40.2 mAh/g at 0.05 C, 0.1 C, 0.2 C, and 0.5 C, respectively, under room temperature, and better capacity retentions of 98% after 300 cycles at 0.05 C.

## 2. Materials and Methods

LCO was bought from Xiamenwuye company (China). Other reagents, which were AR, were brought from Aladdin (China).

The fabrication of LLZO&LNO@LCO using wet-chemical approaches is shown in Figure 1. 1.1 mmol Li_2_CO_3_ and 2 mmol NbCl_5_ were dissolved in water. The 8.4 mmol citrin was added to the solution and stirred at 50 °C for 4 h. Then, 64.3816 g LiCoO_2_ was added to the gel and stirred. Then, the solution was dried at 80 °C to remove the water. LNO@LCO was obtained by calcination at 850 °C for 6 h in air atmosphere. Then, 3.85 mmol Li_2_CO_3_, 1.5 mmol La(NO_3_)_3_•6H_2_O, and 1.5 mmol ZrOCl_2_•8H_2_O were added to the water and stirred. The 25.4 mmol of citrin was added to the solution and stirred at 50 °C for 4 h. Then, 83.7233 g LiCoO_2_ was added to the gel and stirred. The solution was dried at 80 °C to remove the water. LLZO and LNO@LCO was obtained by calcination at 1050 °C for 6 h in air atmosphere.

The contents of elements were analyzed by Agilent 5100VDV. After, 0.1 g LLZO&LNO@LCO was added to 10 mL HNO_3_ (10%). Then, HCl (10%) was added to the solution until LLZO&LNO@LCO dissolved. The solution was diluted to 100 mL. We used SEM (MERlIN VP Compact) to characterize the morphology of LLZO&LNO@LCO at room temperature. XRD was conducted by Empyrean at room temperature with Cu Kα radiation. Raman spectra were recorded using a micro-Raman microscope (In-Via confocal micro, Renishaw) with a 633 nm laser source at room temperature. The porosity of the LCO and LLZO&LNO@LCO was investigated by N_2_ adsorption (Tristar 3020) with an error of less than 5%. Electrochemical measurements were conducted using Solartron 1470E. EIS was performed in an AC field with an amplitude of 10 mV and frequencies ranging from 1 MHz to 1 Hz at room temperature. Charging and discharging cycling was conducted using Land CT3002A at room temperature. The density and volume resistance of LCO and LLZO&LNO@LCO was analyzed by PD-51 at room temperature.

## 3. Results and Discussion

The LCO-bare was synthesized by a conventional high-temperature solid-state method by Xiamenwuye company. LLZO and LNO were coated on the surface of LCO to form LLZO&LNO@LCO by a wet-chemical method. We used ICP to analyze the elements’ contents of LLZO&LNO@LCO. The weight contents of Zr, Nb, La, Li, Co, and O were 1232.72 ppm, 1689.13 ppm, 1873.89 ppm, 7.35%, 59.67%, and 32.50%, respectively, as shown in Table 1. The weight content of the surface-coating materials (LLZO and LNO) was 0.769%, LLZO was 0.503%, and LNO was 0.266%. The molar ratio of Zr:Nb:La = 1:1.348:2.415.

Morphology is very important to the cathode materials. The particle size of LLZO&LNO@LCO is uniform, as shown in Figure 2. LLZO&LNO@LCO was composed of primary particles, as shown in Figure 2. The diameter of the LLZO&LNO@LCO was about 2–8 μm. The surface of LCO-bare was relatively smooth. Additionally, the surface of LLZO&LNO@LCO is very rough, and some minute particles developed on the surface of LCO after coating because of surface-coating layers. The SEM images of the bare and coated particles indicated a smooth surface morphology of the uncoated particles in contrast to the decorated surface of the treated LCO particles. SEM energy-dispersive spectroscopy mapping confirmed the homogeneous distribution of La, Zr, and Nb, as shown in Figure 3; this demonstrates that surface-coating LLZO and LNO were uniform. Compared to Appendix A, LCO particles showed different morphologies without La, Zr, and Nb elements.

X-ray diffraction (XRD) was also conducted to determine the crystallinity of the LCO and LLZO&LNO@LCO to examine the influence of the coating on LCO structure. LCO is the stably layered α-NaFeO_2_ type with *R*$\overset{-}{3}$*m* space group. LLZO&LNO@LCO exhibited strong XRD patterns associated with LCO, as shown in Figure 4A. Most of the diffraction peaks in the patterns were related to layered α-NaFeO_2_ structures corresponding to the space group of *R*$\overset{-}{3}$*m*. LLZO&LNO@LCO and LCO matched well with the PDF#75-0532, without impurity and significant differences. No additional diffraction peaks or secondary phases were observed in the XRD patterns. LLZO&LNO@LCO was also a stably layered α-NaFeO_2_ type with the *R*$\overset{-}{3}$*m* space group. The peak of LLZO and LNO could not be detected obviously because of the low amount of coating materials. No peak shifts were observed, confirming that no significant bulk crystallographic defects were generated in the LCO structure during the coating or annealing step. Clear peak separations of (006)/(012) and (108)/(110) in these four XRD patterns indicated that all as-prepared samples exhibited well-developed crystalline layered structures [25]. Focusing on the (003) and (104) reflections, one can see that both remained unchanged after coating. The *c*-axis and volume change of the LCO lattice are responsible for the evolution of the (003) peak. No shifting of the (003) peak demonstrates that the *c*-axis and volume change of the LCO lattice in LLZO&LNO@LCO had no change. The XRD results indicated that the LLZO and LNO coating had no effect on the structure of LCO, which can be attributed to the low concentration of LLZO and LNO. Crystal lattice constants and d-spacing of LLZO&LNO@LCO and LCO are shown in Table 2.

Raman spectra were used to analyze the coating materials LLZO and LNO. Raman analysis with high surface sensitivity was applied to distinguish the layered structure. Figure 4B shows the Raman spectra of the LCO and LLZO&LNO@LCO. The two characteristic peaks at 477 and 590 cm^−1^ are the characteristic Eg (O-Co-O bending) and A1g (Co-O stretching) Raman peak of the LCO and LLZO&LNO@LCO, suggesting that a certain amount of LCO was maintained after coating. The characteristic peaks at 657 cm^−1^ are the characteristic peaks of LLZO. The characteristic peaks at 173 cm^−1^ are the characteristic peaks of LNO. These suggest that a certain amount of LNO and LLZO was formed after coating.

The porosity of the LCO and LLZO&LNO@LCO was investigated by N_2_ adsorption, which is meaningful for solid-state batteries. The measurements were evaluated using the Brunauer–Emmett–Teller model. The LLZO&LNO@LCO had a higher BET surface of 0.48 m^2^/g, while the LCO had a BET surface of 0.15 m^2^/g because surface-coating materials are highly porous and the surface of LLZO&LNO@LCO is rough. A higher BET surface means more solid–solid contact between LLZO&LNO@LCO and SSEs in solid-state batteries, which is beneficial for Li^+^ transport.

Until now, all of the all-solid-state batteries worked under pressure. Cathode materials may have different discharging capacities under different pressures. The density and volume resistance of LCO and LLZO&SLNO@CLO were analyzed at room temperature, as shown in Figure 5. The density of LLZO&LNO@LCO was lower than that of LCO under the same pressure because LCO has a regular structure and packs tightly, and LLZO&LNO@LCO is rough.

The volume resistance of LCO increased with increasing pressure. The activation barrier for Li hopping was strongly affected by the size of the tetrahedral site and the electrostatic interaction between Li^+^ in that site and the cation in the octahedron that shares a face with it. The size of the tetrahedral site was determined by the *c*-lattice parameter, which had a remarkably strong effect on the activation batteries for Li^+^ migration. The Li slab distance was varied by expanding or contracting the *c*-lattice parameter. Because transition metal (Co)–O bonding is much stiffer than Li–O bonding, the majority of the change in the *c*-lattice parameter was accommodated by the Li slab distance. The *c*-lattice parameter shortened faster than the *a*- (or *b*-) lattice parameter with increasing pressure. The distance of Li–O and Co–O bonding at high pressure made the Li^+^ transport difficult and the volume resistance of LCO increased [26].

Volume resistance of LLZO&LNO@LCO decreased with increasing pressure because of the LLZO and LNO surface-coating layers. Under high pressure, LLZO&LNO@LCO stacked tightly and had more solid–solid contact, and Li^+^ transported faster. The density of LLZO&LNO@LCO was 3.63 mg/cm^3^ at 63.7 MPa, which is similar to the density of LCO at 12.7 MPa. This demonstrates that the lattice parameters of LLZO&LNO@LCO had no obvious change, and the bond distances of Li–O and Co–O remained constant.

The composite cathodes were obtained by mixing cathode powders and LPSC powders at a 7:3 ratio. The typical bulk-type all-solid-state batteries were applied for the electrochemical test. The areal density of cathode materials was 8.9 mg/cm^2^. The Li-In alloys with high anode/SEs interface stability were used as anodes. The SSE membrane was fabricated at 360 MPa for 1 min. The cathode layer was fabricated at 60 MPa for 10 min. The electrochemical performance tests of all-solid-state batteries were conducted in the voltage range between 2.0 and 3.7 V (versus Li^+^/Li-In) using Land CT3002A at room temperature. The capacity-voltage profiles of LLZO&LNO@LCO||LPSC||Li-In and LCO||LPSC||Li-In are shown in Figure 6. The charging and discharging capacities of LLZO&LNO@LCO were evaluated at various *C*-rate from 0.05 C to 0.5 C. The discharging capacity of LLZO&LNO@LCO at 0.05 C was 138.8 mAh/g with a coulombic efficiency of 89.6% at the first cycle. Comparatively, the LLZO&LNO@LCO exhibited a higher specific capacity and coulombic efficiency than LCO at the first three charging and discharging cycles because of the coating layers. The discharging capacities of LCO faded very fast because of the side reaction between the cathode and LPSC. The LLZO and LNO coating layers can suppress the side reaction between cathode and LPSC. Therefore, more Li ions can be inserted back into the structure during discharging. The discharging capacity of LLZO&LNO@LCO is similar to previous results, for example, LCO exhibited an initial capacity of about 130 mAh/h at 0.05 C [27]. Additionally, LLZO&LNO@LCO||LPSC||Li-In had less voltage polarization. LCO||LPSC||Li-In had a higher voltage polarization and fast capacity fade. As the rate increased, charging and discharging capacities of LLZO&LNO@LCO gradually decreased. LLZO&LNO@LCO delivered discharge capacities of at 138.8 mAh/g, 101.8 mAh/g, 60.2 mAh/g, and 40.2 mAh/g at 0.05 C, 0.1 C, 0.2 C, and 0.5 C, respectively, under room temperature, and better capacity retentions of 98% after 300 cycles at 0.05 C. However, the discharging capacity of LCO||LPSC||Li-In decreased quickly from 96 mAh/g to 62.4 mAh/g after three cycles and exhibited much poorer cycling performance at 0.05 C. The fast capacity fade can be attributed to the side reaction between LCO and LPSC, resulting in the incomplete lithium-ion de/intercalation in the crystal lattice.

Electrochemical kinetic properties inside composite cathodes have a significant effect on the cell performance of all-solid-state batteries. The EIS measurements were carried out to further study the mechanism for improved performance of LLZO&LNO@LCO. EIS tests of all-solid-state batteries before and after 20 cycles were conducted as shown in Figure 7. The initial EIS curves contain one semicircle and a tail straight line. The semicircle in the intermediate frequency range represents the charge-transfer impedance and interfacial capacitance between the surface layer and active material. The inclined line in the low frequency band represents the Warburg impedance, which reflects the diffusion ability of lithium ions in inactive material particles. Additionally, they had similar bulk resistances of 13.63 Ω and 17.08 Ω at the intial state, as shown in Table 3. The resistance of anode interfaces of LLZO&LNO@LCO||LPSC||Li-In and LCO||LPSC||Li-In was 36.77 Ω and 32.94 Ω. These demonstrate that the solid-state batteries are fabricated under same conditions. After 20 charging–discharging cycles, The EIS curves contained two semicircles and a tail straight line. The two semicircles in the high-frequency region correspond to the charge-transfer resistance at the cathode interface and anode interfaces. The resistance of anode interfaces is similar to the initial state, demonstrating that Li-In alloy anode is stable. The interfacial resistance of LLZO&LNO@LCO/LPSC was only 70.75 Ω after 20 cycles. However, the interfacial resistance of LCO/LPSC is 185.4 Ω. The interfacial resistance of LCO and LPSC was much larger than that of LLZO&LNO@LCO and LPSC, demonstrating that LLZO&LNO@LCO||LPSC||Li-In had a more stable cathode interface, LLZO and LNO coating treatment could maintain the stability of charge transport kinetics, and LLZO&LNO@LCO was more stable with LPSC than LCO.

To understand the electrochemical reaction kinetics of LLZO&LNO@LCO, the galvanostatic intermittent titration technique (GITT) was employed for LLZO&LNO@LCO. The GITT was employed to track the lithium ion diffusion coefficient (D_Li+_) of all-solid-state batteries in the initial cycle. The diffusion coefficient of Li^+^ was measured by constant current intermittent titration at a constant density of 0.1 C for 20 min after maintaining an open-circuit-potential of 1 h in the voltage range of 2.5–3.7 V (vs. Li^+^/Li). The Li^+^ apparent chemical diffusion coefficient can be calculated using Equation (1):(1)$$D_{{Li}^{+}}=\frac{4}{\pi \tau }{\left(\frac{m_{B}V_{m}}{SM_{B}}\right)}^{2}{\left(\frac{{△E}_{s}}{{△E}_{\tau }}\right)}^{2}\;,\;\tau \ll \frac{L^{2}}{D}$$where D_Li+_ is the diffusion coefficent of Li^+^ in the cathode material and m_B_, V_m_, and M_B_ are the mass, molar volume, and molar weight of the cathode material, respectively. S is the total surface area of the electrode.

The GITT curve of the first cycle is shown in Figure 8. The potential difference of the LLZO&LNO@LCO and LCO at 0.1 C can be calculated. The Li^+^ insertion–extraction process is dominated by diffusion behavior. The diffusion coefficients of Li ion can be calculated by Equation 1. As shown in Figure 8, the electrochemical polarization of LLZO&LNO@LCO was less than LCO. The D_Li+_ of LLZO&LNO@LCO was 1.23 × 10^−9^ cm^2^/s, much higher than that of LCO 4.62 × 10^−10^ cm^2^/s. In a word, the LLZO&LNO@LCO electrode exhibited an excellent rate capability, attributed to its relatively small electrochemical polarization and stable Li^+^ diffusion coefficients [28].

## 4. Conclusions

In conclusion, we first prepared a Li_7_La_3_Zr_2_O_12_-co-LiNbO_3_ coating LCO using wet-chemical approaches to improve the interfacial stability between the cathode and sulfide electrolyte. The influence of the LLZO and LNO coatings on the electrochemical performance, structure, and diffusion kinetics was illustrated by electrochemical measurement, XRD, Raman and kinetic analyses. Structural characterization revealed that the coating process did not destroy the LCO lattices and formed a uniform LLZO&LNO coating layer. LLZO&LNO@LCO had more homogeneous particle size distribution and higher specific surface area than LCO. Additionally, the volume resistance of LLZO&LNO@LCO was lower than LCO because of the Li^+^ conductivity of LLZO and LNO coatings. The high ionic conductivity of LLZO and LNO made it possible for lithium ions to intercalate and deintercalate rapidly on the interface between LLZO&LNO@LCO and LPSC, significantly increasing the diffusion kinetics of Li^+^. The harmful side reaction between LPSC and cathode materials was well alleviated, which improved the stability of LCO. ASSBs containing LLZO&LNO@LCO had a significant improvement in long-term cycling performance due to the high stability of the LLZO and LNO. Additionally, the discharge capacity improved compared with those assembled from uncoated LCO. LLZO&LNO@LCO||LPSC||Li-In delivered discharge capacities of at 138.8 mAh/g, 101.8 mAh/g, 60.2 mAh/g, and 40.2 mAh/g at 0.05 C, 0.1 C, 0.2 C, and 0.5 C, respectively, under room temperature, and better capacity retentions of 98% after 300 cycles at 0.05 C, which is compared to the previous work. The results highlight promising low-cost and scalable cathode material coatings for ASSBs. It is impressive to use LLZO and LNO as the coating material to enhance the electrochemical performance and stability of LCO in solid-state batteries, which provides a new idea for improving the performance of solid-state batteries, and similar materials are expected to be used as coating materials to modify layered cathode materials.

## Figures and Tables

**Figure 1 membranes-13-00216-f001:**
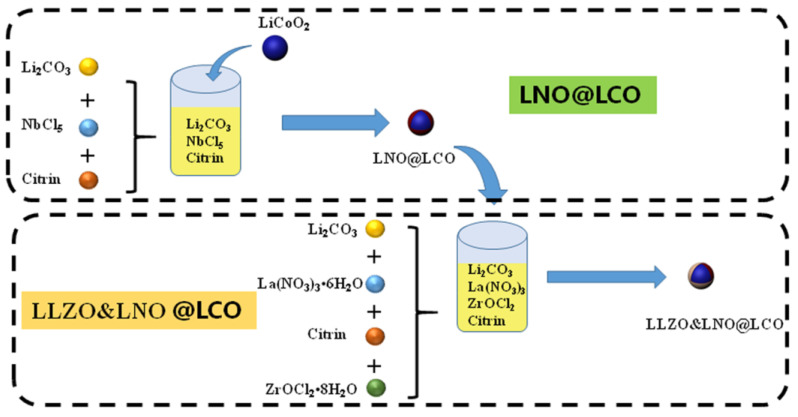
Illustration of the synthesis of LLZO&LNO@LCO.

**Figure 2 membranes-13-00216-f002:**
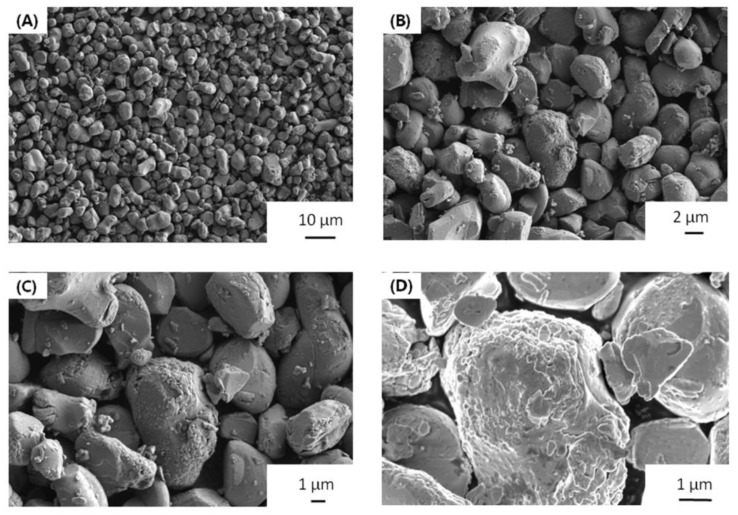
SEM image of LLZO&LNO@LCO with different magnification, 1000 (**A**), 3000 (**B**), 5000 (**C**), 10,000 (**D**).

**Figure 3 membranes-13-00216-f003:**
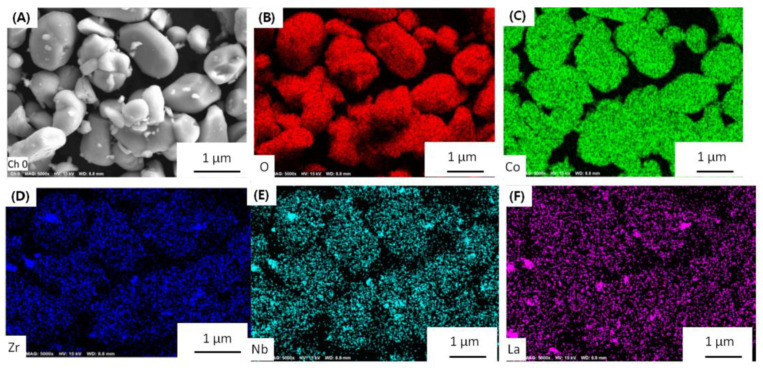
(**A**) selected particles, and elemental O (**B**), Co (**C**), Zr (**D**), Nb (**E**), and La (**F**) mapping of LLZO&LNO@LCO.

**Figure 4 membranes-13-00216-f004:**
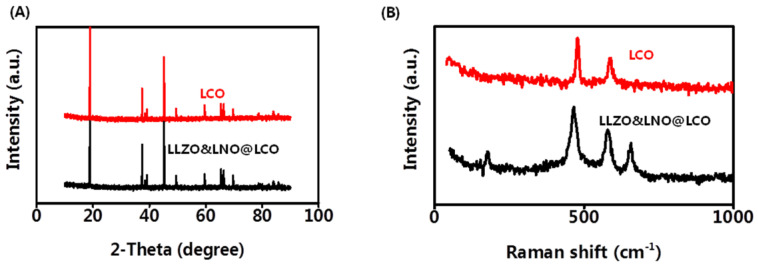
XRD pattern (**A**) and Raman spectra (**B**) of LLZO&LNO@LCO and LCO.

**Figure 5 membranes-13-00216-f005:**
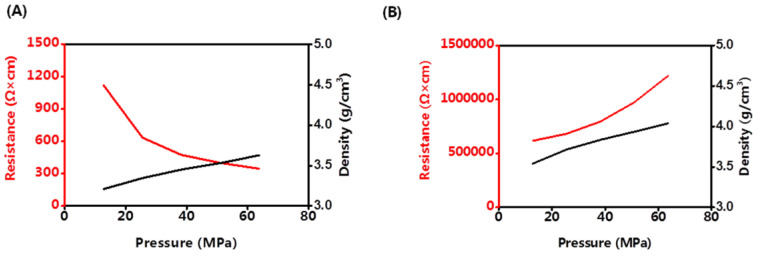
Volume resistance and density of LLZO&LNO@LCO (**A**) and LCO (**B**) under different pressures.

**Figure 6 membranes-13-00216-f006:**
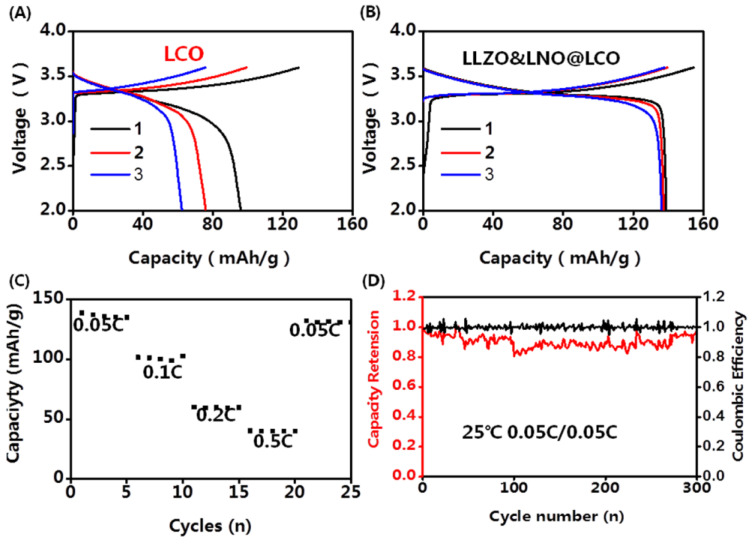
Charging and discharging curve of LCO||LPSC||Li-In (**A**) and LLZO&LNO@LCO||LPSC||Li-In (**B**,**C**) discharging capacity of LLZO&LNO@LCO at a different rate, (**D**) lifecycles of LLZO&LNO@LCO||LPSC||Li-In at 0.05 C.

**Figure 7 membranes-13-00216-f007:**
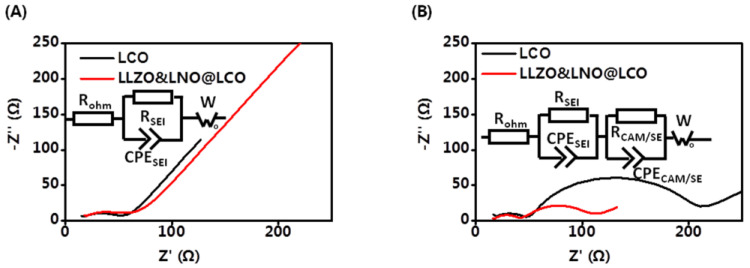
EIS curve of LCO||LPSC||Li-In and LLZO&LNO@LCO||LPSC||Li-In before (**A**) and after 20 cycles (**B**).

**Figure 8 membranes-13-00216-f008:**
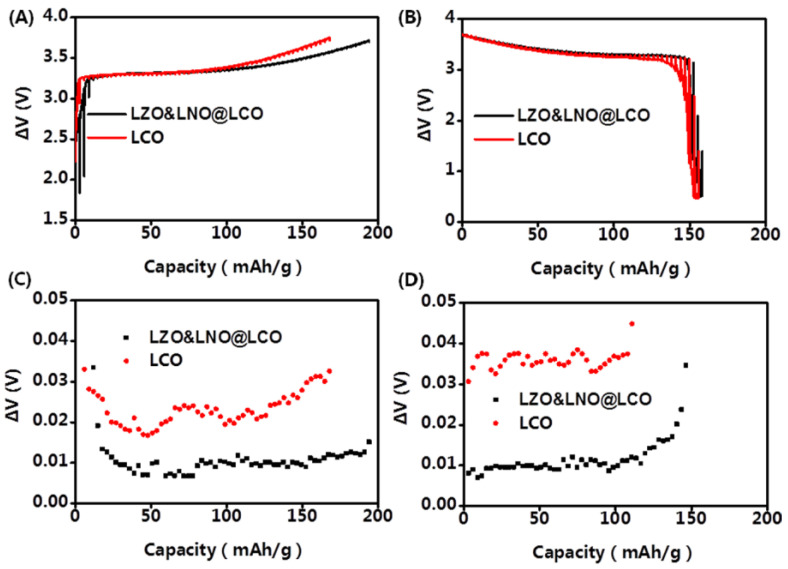
Charging GITT profiles (**A**) and discharging GITT profiles (**B**) of LLZO&LNO@LCO and LCO samples measured at the first cycle. Charging polarization voltage profiles (**C**) and discharging polarization GITT profiles (**D**) of LLZO&LNO@LCO and LCO samples measured at the first cycle.

**Table 1 membranes-13-00216-t001:** Element composition and weight concentration of LLZO&LNO@LCO and LCO.

wt/wt	Li (%)	Co (%)	Zr (ppm)	Nb (ppm)	La (ppm)
LLZO&LNO@LCO	7.35	59.67	1232.72	1689.13	1873.89
LCO	7.22	60.18	/	/	/

**Table 2 membranes-13-00216-t002:** Crystal lattice constants and d-spacing of LLZO&LNO@LCO and LCO.

Materials	a (Å)	b (Å)	c (Å)	α (°)	β (°)	γ (°)
LLZO&LNO@LCO	2.8158	2.8158	14.0511	90	90	120
LCO	2.8127	2.8127	14.0815	90	90	120

**Table 3 membranes-13-00216-t003:** Impedance parameters of LCO||LPSC||Li-In and LLZO&LNO@LCO||LPSC||Li-In before (A) and after 20 cycles (B).

Cathode		R_ohm_ (Ω)	R_SEI_ (Ω)	R_CAM/SE_ (Ω)
LLZO&LNO@LCO	Initial	13.63	36.77	/
	After 20 cycles	15.22	36.41	70.75
LCO	Initial	17.08	32.94	/
	After 20 cycles	16.95	38.15	185.4

## Data Availability

Not applicable.

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
