# Peer review of "Li7La3Zr2O12-co-LiNbO3 Surface Modification Improves the Interface Stability between Cathode and Sulfide Solid-State Electrolyte in All-Solid-State Batteries"

_membranes, 2023, doi:10.3390/membranes13020216_

Round 1

Reviewer 1 Report

This paper is complete in the sense that its give overall scenario of membrane in energy devicesIts written well and explain everything. The results and discussion are well correlated.

Hence it is recommended to accept as such

Author Response

Dear Editors:

On behalf of my co-authors, we appreciate the editor and reviewers very much for their comments and suggestions on our manuscript entitled “Li7La3Zr2O12-co-LiNbO3 Surface Modification improves Interface Stability between Cathode and Sulfide Solid-State Electrolyte in All-Solid-State Batteries”.

With the rapid development of energy storage and electric vehicles, thiophosphate-based ASSBs are considered the most promising power source. In order to commercialize ASSBs, the interfacial problem between high-voltage cathode active materials and thiophosphate-based solid-state electrolytes needs to be solved in a simple, effective way. Surface coatings are considered the most promising approach to solving the interfacial problem because surface coatings could prevent direct physical contact between cathode active materials and thiophosphate-based solid-state electrolytes. In this work, LLZO and LNO coatings for LCO is fabricated by in-situ interfacial growth of two high Li+ conductive oxide electrolytes on the LCO surface and tested for thiophosphate-based ASSBs. The coatings are obtained from a two-step traditional sol-gel coatings process, the inner coatings are LNO, and the surface coatings are LLZO. Electrochemical evaluations confirm that the two-layer coatings are beneficial for ASSBs. ASSBs containing LLZO&LNO@LCO significantly improve long-term cycling performance and discharge capacity compared to those assembled from uncoated LCO. LLZO&LNO@LCO||LPSC||Li-In delivers discharge capacities of 138.8 mAh/g, 101.8 mAh/g, 60.2 mAh/g, 40.2 mAh/g at 0.05 C, 0.1 C, 0.2 C, 0.5 C under room temperature respectively, and better capacity retentions of 98% after 300 cycles at 0.05 C. The results highlight promising low-cost and scalable cathode materials coatings for ASSBs.

We have studied the reviewers’ comments carefully and have revised. We have tried our best to revise our manuscript according to the comments. Attached please find the revised version, which we would like to resubmit for your kind consideration.

We would like to express our great appreciation to you and reviewers for your comments on our paper. Looking forward to hearing from you.

Thank you and best regards.

Yours sincerely,

Shusen Kang

E-mail: kshusen@163.com

Responds to the reviewer’s comments:

Reviewer: 1

This paper is complete in the sense that its give overall scenario of membrane in energy devicesIts written well and explain everything. The results and discussion are well correlated.

Hence it is recommended to accept as such.

Response:

Thanks for your comments.

Reviewer 2 Report

Manuscript entitled "Li7La3Zr2O12-co-LiNbO3 Surface Modification improves Interface Stability between Cathode and Sulfide Solid-State Electrolyte in All-Solid-State Batteries" is in agreement with the general direction of the Membranes journal publications. However, it is worth considering comments on the manuscript before submitting it to the journal Membranes.

1. The abbreviations and acronyms to be introduced should be deciphered from the text of the paper.

2. On page 2 the sentence in paragraph 2 is broken.

3. The country of the chemical factory must be written down. Specify the purity of each reagent.

4. Since the synthesis of LNO@LCO and LLZO&LNO@LCO was obtained by co-evaporation and roasting of the initial components, it is necessary to prove the formation and existence of all described phases in the composition.

5. Figure 1 should be located in section 2. Materials and Methods.

6. Decipher the ICP method of analysis, the equipment used for analysis. Describe the approach to material analysis by inductively coupled plasma mass spectrometry, method of sample preparation of powders, their dissolution, etc.

7. Is it possible to make an in-depth analysis of the compounds from ICP analysis? It is not clear from the text what is the elemental content of Zr, Nb, La, Li, Co, and O in ppm or %.  Are the percentages indicated in atom% or mol%? If they are atomic percentages, then for the Li2CoO2 compound the Li/Co ratio is about 1/1, which does not correspond to the formula content. It is necessary to describe this part in more detail in order to evaluate the resulting powder composition.

8. The methodological part of the paper is poor, it is necessary to specify all the equipment and techniques used. In the methods it is necessary to specify the type of microscope for SEM imaging and attachment for X-ray analysis; diffractometer on which the X-ray patterns were obtained; instrument for specific surface measurement; resistance measurement technique and equipment for experiments; methodology of galvanostatic intermittent titration and equipment.

9. It looks like the LLZO&LNO@LCO in figure 2 is powder. Is this correct? In the text there is a description as coatings.

10. There is no reference in the text to Figures S1 and S2 from the supplementary material. A description needs to be added to the manuscript text.

11. Crystalline lattice parameters and unit cell volume should be given to prove that there is no change in the composition of the cathode. Figures 4 and S3 should be combined for better visualization and understanding of the manuscript text.

12. As the specific surface area is low for both LLZO&LNO@LCO and LCO, it is necessary to specify the measurement error of the specific characteristics in order to estimate their difference.

13. On page 5 the density behaviour of the materials is described and the pressure dependencies of the densities are given, but there is no explanation of how the densities were determined.

14. Since the specific surface area is low for both LLZO&LNO@LCO and LCO, it is necessary to specify the measurement error of the specific characteristics in order to estimate their difference.

15. What difference do you make in the terminology of volume resistance and power resistance? The terminology in the text and in the description of the figure is different (Fig. 5). There is also no reference to Fig. 5 in the text.

16. Figure 6d is not discussed in the text.

17. It is necessary to explain how the diffusion coefficient of lithium ions was calculated and how the stability of the diffusion characteristics for the LLZO&LNO@LCO electrode was evaluated.

Author Response

Dear Editors:

On behalf of my co-authors, we appreciate the editor and reviewers very much for their comments and suggestions on our manuscript entitled “Li7La3Zr2O12-co-LiNbO3 Surface Modification improves Interface Stability between Cathode and Sulfide Solid-State Electrolyte in All-Solid-State Batteries”.

With the rapid development of energy storage and electric vehicles, thiophosphate-based ASSBs are considered the most promising power source. In order to commercialize ASSBs, the interfacial problem between high-voltage cathode active materials and thiophosphate-based solid-state electrolytes needs to be solved in a simple, effective way. Surface coatings are considered the most promising approach to solving the interfacial problem because surface coatings could prevent direct physical contact between cathode active materials and thiophosphate-based solid-state electrolytes. In this work, LLZO and LNO coatings for LCO is fabricated by in-situ interfacial growth of two high Li+ conductive oxide electrolytes on the LCO surface and tested for thiophosphate-based ASSBs. The coatings are obtained from a two-step traditional sol-gel coatings process, the inner coatings are LNO, and the surface coatings are LLZO. Electrochemical evaluations confirm that the two-layer coatings are beneficial for ASSBs. ASSBs containing LLZO&LNO@LCO significantly improve long-term cycling performance and discharge capacity compared to those assembled from uncoated LCO. LLZO&LNO@LCO||LPSC||Li-In delivers discharge capacities of 138.8 mAh/g, 101.8 mAh/g, 60.2 mAh/g, 40.2 mAh/g at 0.05 C, 0.1 C, 0.2 C, 0.5 C under room temperature respectively, and better capacity retentions of 98% after 300 cycles at 0.05 C. The results highlight promising low-cost and scalable cathode materials coatings for ASSBs.

We have studied the reviewers’ comments carefully and have revised. We have tried our best to revise our manuscript according to the comments. Attached please find the revised version, which we would like to resubmit for your kind consideration.

We would like to express our great appreciation to you and reviewers for your comments on our paper. Looking forward to hearing from you.

Thank you and best regards.

Yours sincerely,

Shusen Kang

E-mail: kshusen@163.com

Responds to the reviewer’s comments:

Reviewer: 2

Manuscript entitled "Li7La3Zr2O12-co-LiNbO3 Surface Modification improves Interface Stability between Cathode and Sulfide Solid-State Electrolyte in All-Solid-State Batteries" is in agreement with the general direction of the Membranes journal publications. However, it is worth considering comments on the manuscript before submitting it to the journal Membranes.

Response:

Thanks for your comments.

  1. The abbreviations and acronyms to be introduced should be deciphered from the text of the paper.

Response:

I mark all the abbreviations and acronyms from the text of paper, for example,

Li7La3Zr2O12 (LLZO)

LiNbO3 (LNO)

LiCoO2 (LCO)

LLZO-co-LNO coatings LiCoO2 (LLZO&LNO@LCO)

Li10GeP2S12 (LGPS)

Li6PS5Cl (LPSC)

  1. On page 2 the sentence in paragraph 2 is broken.

Response:

The broken sentence has been corrected.

  1. The country of the chemical factory must be written down. Specify the purity of each reagent.

Response:

The country of the chemical factory and purity of each reagent are written down.

  1. Since the synthesis of LNO@LCO and LLZO&LNO@LCO was obtained by co-evaporation and roasting of the initial components, it is necessary to prove the formation and existence of all described phases in the composition.

Response:

The EDS-mapping of LLZO&LNO@LCO has been proved the existence of Nb, La, Zr. XRD and Roman have proven the LNO and LLZO phase.

  1. Figure 1 should be located in section 2. Materials and Methods.

Response:

The location of Figure 1 has been changed to section 2.

  1. Decipher the ICP method of analysis, the equipment used for analysis. Describe the approach to material analysis by inductively coupled plasma mass spectrometry, method of sample preparation of powders, their dissolution, etc.

Response:

The ICP was conducted as following steps:

The experiment was conducted by Agilent 5100VDV. 0.1 g LLZO&LNO@LCO added to 10mL HNO3 (10%). Then HCl (10%) added to the solution until LLZO&LNO@LCO dissolves. The solution was diluted to 100 mL, then the solution was analysized.

  1. Is it possible to make an in-depth analysis of the compounds from ICP analysis? It is not clear from the text what is the elemental content of Zr, Nb, La, Li, Co, and O in ppm or %.  Are the percentages indicated in atom% or mol%? If they are atomic percentages, then for the Li2CoO2 compound the Li/Co ratio is about 1/1, which does not correspond to the formula content. It is necessary to describe this part in more detail in order to evaluate the resulting powder composition.

Response:

the elemental content of Zr, Nb, La, Li, Co, and O is weight percentages (wt%). This has been highlihgt in the articles.

And the LLZO and LNO weight ratio was analysised.

  1. The methodological part of the paper is poor, it is necessary to specify all the equipment and techniques used. In the methods it is necessary to specify the type of microscope for SEM imaging and attachment for X-ray analysis; diffractometer on which the X-ray patterns were obtained; instrument for specific surface measurement; resistance measurement technique and equipment for experiments; methodology of galvanostatic intermittent titration and equipment.

Response:

The methodogical part was re-written.

  1. It looks like the LLZO&LNO@LCO in figure 2 is powder. Is this correct? In the text there is a description as coatings.

Response:

Yes, the SEM image shows the LLZO&LNO@LCO particle morphology. LLZO and LNO is the coatings.

  1. There is no reference in the text to Figures S1 and S2 from the supplementary material. A description needs to be added to the manuscript text.

Response:

The necessary was added.

  1. Crystalline lattice parameters and unit cell volume should be given to prove that there is no change in the composition of the cathode. Figures 4 and S3 should be combined for better visualization and understanding of the manuscript text.

Response:

This part was re-written. What is more, Figure S3 has been deleted because the noise is too high. We used the Raman to analysize the coating layers.

  1. As the specific surface area is low for both LLZO&LNO@LCO and LCO, it is necessary to specify the measurement error of the specific characteristics in order to estimate their difference.

Response:

The measurement error is less 0.5%.

  1. On page 5 the density behaviour of the materials is described and the pressure dependencies of the densities are given, but there is no explanation of how the densities were determined.

Response:

The density and volume resistance of LCO and LLZO&SLNO@CLO was analysized by PD-51 at room temperature with measurement error less 0.5%.

  1. Since the specific surface area is low for both LLZO&LNO@LCO and LCO, it is necessary to specify the measurement error of the specific characteristics in order to estimate their difference.

Response:

The porosity of the LCO and LLZO&LNO@LCO was investigated by N2 adsorption (Tristar 3020) with an error of less than 5%

  1. What difference do you make in the terminology of volume resistance and power resistance? The terminology in the text and in the description of the figure is different (Fig. 5). There is also no reference to Fig. 5 in the text.

Response:

They are same in this work. And power resistance has been changed into volume resistance. And Figure 5 is refered now.

  1. Figure 6d is not discussed in the text.

Response:

The Figure 6 is disscussed again.

  1. It is necessary to explain how the diffusion coefficient of lithium ions was calculated and how the stability of the diffusion characteristics for the LLZO&LNO@LCO electrode was evaluated.

Response:

The calculation of diffusion coefficient of lithium ions has been demonstrated in article carefully.

Reviewer 3 Report

The article entitled "Li7La3Zr2O12-co-LiNbO3 Surface Modification improves Interface Stability between Cathode and Sulfide Solid-State Electrolyte in All-Solid-State Batteries" presents the surface coatings for the interface stability improvement of ASSBs. 

Coating the cathode active material to improve the interfacial stability between the cathode and sulfide solid-state electrolyte is a relevant task. However, the article's novelty is not obvious. The question remains: What is the scientific or practical novelty of this research compared to other similar works? 

The reviewer’s comments are shown below. 

(1) Introduction, lines 85-88: the authors write that they prepared the Li7La3Zr2O12 and LiNiO3 coating, however, the results of research describe the LiNbO3.

The same mistake is in Conclusions.

(2) Lines 110-111: 'The 25.4 mmol of 8 citrin'... What does it mean?

(3) In what atmosphere was the calcination carried out?

(4) There is no description of methods in paragraph 2 Materials and Methods. It is critical to include sufficient information about the experimental conditions of characterization methods.

(5) Fig.1 refers to paragraph 2 Materials and Methods. 

(6) The meaning of all abbreviations should be explained at the first mention. For example, solid-state electrolyte (SSE) or Li6PS5Cl (LPSC) etc.

(7) Lines 125-126: ‘The particle size of LLZO&LNO@LCO is uniform as shown in figure 2.’ For this claim, the particle size distribution analysis should be done.

(8) Fig. 4: The x-axis label is wrong. The units for the y-axis should be added.

(9) Lines 145-146: ‘We enlarge the XRD of LLZO&LNO@LCO as shown in figure S3. XRD peaks of LLZO and LNO could be seen.’ Fig. S3 has low quality with strong noise. There are no diffraction peaks of coatings on the XRD pattern. The XRD analysis should be performed using settings provided high quality of the XRD pattern.

(10) The coatings’ characterization should be performed. To determine the crystalline structure of the coatings on LCO, Raman spectroscopy needs to be carried out. Or obtaining the chemical composition of the coatings can be provided via XPS analysis.

(11) Lines 159-160: ‘The density of LLZO&LNO@LCO is lower than that of LCO under the same pressure because LCO has a regular structure and packs tightly, and LLZO&LNO@LCO is rough.’ How was the density determined?

(12) The authors write about the dependence of Li-ion hopping on the crystal lattice parameters (lines 164-170), and that the c-lattice parameter doesn’t change for LLZO&LNO@LCO (lines 176-177). The lattice parameters should be calculated from XRD patterns and added to the relevant paragraph.

(13) There is no reference to Fig. 5 in text.

(14) The batteries' parameters of the electrochemical testing should be compared with other similar research. Are the obtained results, such as the specific capacity and capacity retention, higher or comparable to those of developing analogues at the same C-rate? Please give examples. 

(15) Provide the electrical circuits for the Nyquist's diagrams (Fig.7) and describe the processes.

(16) The Conclusions should be revised in accordance with the above-mentioned comments and give significant results.

The reviewer cannot recommend this article for publication in Membranes.

Author Response

Dear Editors:

On behalf of my co-authors, we appreciate the editor and reviewers very much for their comments and suggestions on our manuscript entitled “Li7La3Zr2O12-co-LiNbO3 Surface Modification improves Interface Stability between Cathode and Sulfide Solid-State Electrolyte in All-Solid-State Batteries”.

With the rapid development of energy storage and electric vehicles, thiophosphate-based ASSBs are considered the most promising power source. In order to commercialize ASSBs, the interfacial problem between high-voltage cathode active materials and thiophosphate-based solid-state electrolytes needs to be solved in a simple, effective way. Surface coatings are considered the most promising approach to solving the interfacial problem because surface coatings could prevent direct physical contact between cathode active materials and thiophosphate-based solid-state electrolytes. In this work, LLZO and LNO coatings for LCO is fabricated by in-situ interfacial growth of two high Li+ conductive oxide electrolytes on the LCO surface and tested for thiophosphate-based ASSBs. The coatings are obtained from a two-step traditional sol-gel coatings process, the inner coatings are LNO, and the surface coatings are LLZO. Electrochemical evaluations confirm that the two-layer coatings are beneficial for ASSBs. ASSBs containing LLZO&LNO@LCO significantly improve long-term cycling performance and discharge capacity compared to those assembled from uncoated LCO. LLZO&LNO@LCO||LPSC||Li-In delivers discharge capacities of 138.8 mAh/g, 101.8 mAh/g, 60.2 mAh/g, 40.2 mAh/g at 0.05 C, 0.1 C, 0.2 C, 0.5 C under room temperature respectively, and better capacity retentions of 98% after 300 cycles at 0.05 C. The results highlight promising low-cost and scalable cathode materials coatings for ASSBs.

We have studied the reviewers’ comments carefully and have revised. We have tried our best to revise our manuscript according to the comments. Attached please find the revised version, which we would like to resubmit for your kind consideration.

We would like to express our great appreciation to you and reviewers for your comments on our paper. Looking forward to hearing from you.

Thank you and best regards.

Yours sincerely,

Shusen Kang

E-mail: kshusen@163.com

Responds to the reviewer’s comments:

Reviewer: 3

The article entitled "Li7La3Zr2O12-co-LiNbO3 Surface Modification improves Interface Stability between Cathode and Sulfide Solid-State Electrolyte in All-Solid-State Batteries" presents the surface coatings for the interface stability improvement of ASSBs.

Coating the cathode active material to improve the interfacial stability between the cathode and sulfide solid-state electrolyte is a relevant task. However, the article's novelty is not obvious. The question remains: What is the scientific or practical novelty of this research compared to other similar works?

Response:

Coating the cathode active material to improve the interfacial stability between the cathode and sulfide solid-state electrolyte is an important tast to ASSBs. However, cathode materials in previous articles usually contain one coating-layers, which could not a very satisfactory results.

In this work, we LLZO and LNO coatings for LCO is fabricated by in-situ interfacial growth of two high Li+ conductive oxide electrolytes on the LCO surface and tested for thiophosphate-based ASSBs. The coatings are obtained from a two-step traditional sol-gel coatings process, the inner coatings are LNO, and the surface coatings are LLZO. ASSBs containing LLZO-co-LNO coatings LiCoO2 (LLZO&LNO@LCO) significantly improve long-term cycling performance and discharge capacity compared to those assembled from uncoated LCO. And discharge capacity, compared to those assembled from uncoated LCO. LLZO&LNO@LCO||LPSC||Li-In delivers discharge capacities of at 138.8 mAh/g, 101.8 mAh/g, 60.2 mAh/g, 40.2 mAh/g at 0.05 C, 0.1 C, 0.2 C, 0.5 C under room temperature respectively, and better capacity retentions of 98% after 300 cycles at 0.05 C, which is compared to the privious work. The results highlight promising low-cost and scalable cathode materials coatings for ASSBs.

The reviewer’s comments are shown below.

(1) Introduction, lines 85-88: the authors write that they prepared the Li7La3Zr2O12 and LiNiO3 coating, however, the results of research describe the LiNbO3. The same mistake is in Conclusions.

Response:

I correct the mistakes carefully.

(2) Lines 110-111: 'The 25.4 mmol of 8 citrin'... What does it mean?

Response:

 I correct the mistakes carefully.

(3) In what atmosphere was the calcination carried out?

Response:

LNO@LCO was got by calcined at 850 ℃ for 6 h in air atmosphere

LLZO&LNO@LCO was got by calcined at 1050 ℃ for 6 h in air atmosphere.

(4) There is no description of methods in paragraph 2 Materials and Methods. It is critical to include sufficient information about the experimental conditions of characterization methods.

Response:

Description of methods is shown in the results and discussion. The sufficient information about the experimental conditions of characterization methods is bofore the results.

(5) Fig.1 refers to paragraph 2 Materials and Methods.

Response:

The location of Figure 1 has been changed to section 2.

 (6) The meaning of all abbreviations should be explained at the first mention. For example, solid-state electrolyte (SSE) or Li6PS5Cl (LPSC) etc.

Response:

I mark all the abbreviations and acronyms from the text of paper, for example,

Li7La3Zr2O12 (LLZO)

LiNbO3 (LNO)

LiCoO2 (LCO)

LLZO-co-LNO coatings LiCoO2 (LLZO&LNO@LCO)

Li10GeP2S12 (LGPS)

Li6PS5Cl (LPSC)

(7) Lines 125-126: ‘The particle size of LLZO&LNO@LCO is uniform as shown in figure 2.’ For this claim, the particle size distribution analysis should be done.

Response:

 The particle size distribution analysis is shown in figure S3 in SI

Figure S3.Particle size distribution of LLZO&LNO@LCO and LCO.

 (8) Fig. 4: The x-axis label is wrong. The units for the y-axis should be added.

Response:

I correct the mistakes carefully.

(9) Lines 145-146: ‘We enlarge the XRD of LLZO&LNO@LCO as shown in figure S3. XRD peaks of LLZO and LNO could be seen.’ Fig. S3 has low quality with strong noise. There are no diffraction peaks of coatings on the XRD pattern. The XRD analysis should be performed using settings provided high quality of the XRD pattern.

Response:

This part was re-written. What is more, Figure S3 has been deleted because the noise is too high. We used the Raman to analysize the coating layers.

 (10) The coatings’ characterization should be performed. To determine the crystalline structure of the coatings on LCO, Raman spectroscopy needs to be carried out. Or obtaining the chemical composition of the coatings can be provided via XPS analysis.

Response:

We used Raman spectroscopy to determine the crystalline structure of the coating on LCO.

(11) Lines 159-160: ‘The density of LLZO&LNO@LCO is lower than that of LCO under the same pressure because LCO has a regular structure and packs tightly, and LLZO&LNO@LCO is rough.’ How was the density determined?

Response:

The density and volume resistance of LCO and LLZO&SLNO@CLO was analysized by PD-51 at room temperature with measurement error less 0.5%.

 (12) The authors write about the dependence of Li-ion hopping on the crystal lattice parameters (lines 164-170), and that the c-lattice parameter doesn’t change for LLZO&LNO@LCO (lines 176-177). The lattice parameters should be calculated from XRD patterns and added to the relevant paragraph.

Response:

The density is measured by the PD-51 at room temperature. This instrument could be applied to measure the XRD of powder at different pressure. And we could find appropriate instruments to finish this experiment.

What’s more, reference 27 has demonstrated that influence of pressure on the volume resistance, and we had citing this literature. The low

(13) There is no reference to Fig. 5 in text.

Response:

 And Figure 5 is refered now.

(14) The batteries' parameters of the electrochemical testing should be compared with other similar research. Are the obtained results, such as the specific capacity and capacity retention, higher or comparable to those of developing analogues at the same C-rate? Please give examples.

Response:

The discharging capacity of LLZO&LNO@LCO at 0.05 C is  138.8 mAh/g with a coulombic efficiency of 89.6% at first cycle. The discharging capacity of LLZO&LNO@LCO is comparable to silimar results, for example, LCO with 15% LIC exhibits a initial capacity 131.7 mAh/h at 0.1 C[23].

(15) Provide the electrical circuits for the Nyquist's diagrams (Fig.7) and describe the processes.

Response:

The Nyquist’s diagrams is provided in figure 7.

(16) The Conclusions should be revised in accordance with the above-mentioned comments and give significant results.

Response:

This part was re-written.

Round 2

Reviewer 2 Report

All comments were taken into account. The revised version of the manuscript looks much better. The manuscript may be published

Author Response

Dear Editors:

On behalf of my co-authors, we appreciate the editor and reviewers very much for their comments and suggestions on our manuscript entitled “Li7La3Zr2O12-co-LiNbO3 Surface Modification improves Interface Stability between Cathode and Sulfide Solid-State Electrolyte in All-Solid-State Batteries”.

With the rapid development of energy storage and electric vehicles, thiophosphate-based ASSBs are considered the most promising power source. In order to commercialize ASSBs, the interfacial problem between high-voltage cathode active materials and thiophosphate-based solid-state electrolytes needs to be solved in a simple, effective way. Surface coatings are considered the most promising approach to solving the interfacial problem because surface coatings could prevent direct physical contact between cathode active materials and thiophosphate-based solid-state electrolytes. In this work, LLZO and LNO coatings for LCO is fabricated by in-situ interfacial growth of two high Li+ conductive oxide electrolytes on the LCO surface and tested for thiophosphate-based ASSBs. The coatings are obtained from a two-step traditional sol-gel coatings process, the inner coatings are LNO, and the surface coatings are LLZO. Electrochemical evaluations confirm that the two-layer coatings are beneficial for ASSBs. ASSBs containing LLZO&LNO@LCO significantly improve long-term cycling performance and discharge capacity compared to those assembled from uncoated LCO. LLZO&LNO@LCO||LPSC||Li-In delivers discharge capacities of 138.8 mAh/g, 101.8 mAh/g, 60.2 mAh/g, 40.2 mAh/g at 0.05 C, 0.1 C, 0.2 C, 0.5 C under room temperature respectively, and better capacity retentions of 98% after 300 cycles at 0.05 C. The results highlight promising low-cost and scalable cathode materials coatings for ASSBs.

We have studied the reviewers’ comments carefully and have revised. We have tried our best to revise our manuscript according to the comments. Attached please find the revised version, which we would like to resubmit for your kind consideration.

We would like to express our great appreciation to you and reviewers for your comments on our paper. Looking forward to hearing from you.

Thank you and best regards.

Yours sincerely,

Shusen Kang

E-mail: kshusen@163.com

Responds to the reviewer’s comments:

Reviewer 2:

All comments were taken into account. The revised version of the manuscript looks much better. The manuscript may be published

Response:

Thanks for your comments.

Reviewer 3 Report

The reviewer got acquainted with the revised article. However, I still have some comments:

  1. Description of all methods should be presented in the paragraph 2 Materials and Methods, not in the Results and Discussion. 

  2. Line 137: 'XRD was conducted by Empyrean at temperature.' Room temperature? What kind of radiation was used to record XRD patterns?

  3. Authors' Response: 'The particle size distribution analysis is shown in figure S3 in SI. Figure S3. Particle size distribution of LLZO&LNO@LCO and LCO.'

In the MDPI dashboard, the SI old version is loaded. There is no particle size distribution analysis.

  1. Line 169: 'LLZO&SLNO@CLO was analysized by PD-51'. Misspelling of the word 'analyzed'.

  2. The crystal lattice constants and d-spacing could be calculated from XRD patterns in addition to the density measurements to demonstrate no lattice changes for LLZO&LNO@LCO.

  3. Lines 203-206, the comparison of the discharging capacity of LLZO&LNO@LCO at 0.05C with the discharging capacity of LCO with 15% LIC at 0.1С is incorrect. It is needed to use the data obtained at the same test conditions, for example, at the same C-rate.

  4. The equivalent circuits should be revised. The ohmic resistance and Warburg element are missed. It is obvious that the equivalent circuits are different before and after 20 cycles. Impedance parameters obtained by fitting Nyquist plots can be presented as well. The following literature may help the authors to improve the EIS analysis:

[1] 10.1021/acsami.2c09841

[2] 10.1016/j.cap.2022.06.004

[3] 10.33961/jecst.2019.00528

[4] 10.1016/j.jpowsour.2022.232450

After major revision and responding to comments, the article could be published.

Author Response

Dear Editors:

On behalf of my co-authors, we appreciate the editor and reviewers very much for their comments and suggestions on our manuscript entitled “Li7La3Zr2O12-co-LiNbO3 Surface Modification improves Interface Stability between Cathode and Sulfide Solid-State Electrolyte in All-Solid-State Batteries”.

With the rapid development of energy storage and electric vehicles, thiophosphate-based ASSBs are considered the most promising power source. In order to commercialize ASSBs, the interfacial problem between high-voltage cathode active materials and thiophosphate-based solid-state electrolytes needs to be solved in a simple, effective way. Surface coatings are considered the most promising approach to solving the interfacial problem because surface coatings could prevent direct physical contact between cathode active materials and thiophosphate-based solid-state electrolytes. In this work, LLZO and LNO coatings for LCO is fabricated by in-situ interfacial growth of two high Li+ conductive oxide electrolytes on the LCO surface and tested for thiophosphate-based ASSBs. The coatings are obtained from a two-step traditional sol-gel coatings process, the inner coatings are LNO, and the surface coatings are LLZO. Electrochemical evaluations confirm that the two-layer coatings are beneficial for ASSBs. ASSBs containing LLZO&LNO@LCO significantly improve long-term cycling performance and discharge capacity compared to those assembled from uncoated LCO. LLZO&LNO@LCO||LPSC||Li-In delivers discharge capacities of 138.8 mAh/g, 101.8 mAh/g, 60.2 mAh/g, 40.2 mAh/g at 0.05 C, 0.1 C, 0.2 C, 0.5 C under room temperature respectively, and better capacity retentions of 98% after 300 cycles at 0.05 C. The results highlight promising low-cost and scalable cathode materials coatings for ASSBs.

We have studied the reviewers’ comments carefully and have revised. We have tried our best to revise our manuscript according to the comments. Attached please find the revised version, which we would like to resubmit for your kind consideration.

We would like to express our great appreciation to you and reviewers for your comments on our paper. Looking forward to hearing from you.

Thank you and best regards.

Yours sincerely,

Shusen Kang

E-mail: kshusen@163.com

Responds to the reviewer’s comments:

Reviewer: 3

The reviewer got acquainted with the revised article. However, I still have some comments:

Response:

Thanks for your comments.

  1. Description of all methods should be presented in the paragraph 2 Materials and Methods, not in the Results and Discussion.

Response:

I correct this question.

  1. Line 137: 'XRD was conducted by Empyrean at temperature.' Room temperature? What kind of radiation was used to record XRD patterns?

Response:

I correct this question.

  1. Authors' Response: 'The particle size distribution analysis is shown in figure S3 in SI. Figure S3. Particle size distribution of LLZO&LNO@LCO and LCO.' In the MDPI dashboard, the SI old version is loaded. There is no particle size distribution analysis.

Response:

I correct this question and reupload the supporting information.

  1. Line 169: 'LLZO&SLNO@CLO was analysized by PD-51'. Misspelling of the word 'analyzed'.

Response:

I correct this question.

  1. The crystal lattice constants and d-spacing could be calculated from XRD patterns in addition to the density measurements to demonstrate no lattice changes for LLZO&LNO@LCO.

Response:

The crystal lattice constants and d-spacing before and after coating are presented in table 2.

  1. Lines 203-206, the comparison of the discharging capacity of LLZO&LNO@LCO at 0.05C with the discharging capacity of LCO with 15% LIC at 0.1С is incorrect. It is needed to use the data obtained at the same test conditions, for example, at the same C-rate.

Response:

The comparison of discharging of LCO at 0.05C was similar to our work. The reference is corrected.

  1. The equivalent circuits should be revised. The ohmic resistance and Warburg element are missed. It is obvious that the equivalent circuits are different before and after 20 cycles. Impedance parameters obtained by fitting Nyquist plots can be presented as well. The following literature may help the authors to improve the EIS analysis:

[1] 10.1021/acsami.2c09841

[2] 10.1016/j.cap.2022.06.004

[3] 10.33961/jecst.2019.00528

[4] 10.1016/j.jpowsour.2022.232450

Response:

The equivalent circuits is revised, the ohmic resistance and Warburg element are supplemented. Impedance parameters obtained by fitting Nyquist are presented on table 3.

Round 3

Reviewer 3 Report

The revised version of the manuscript can be accepted after minor corrections below.

  1. Methods description was moved to 2 Materials and Methods, but the same text is still in the Results and Discussion. It should be deleted from the Results and Discussion.

  2. Table 2: the first column is entitled 'phase'. It is incorrect. LLZO&LNO@LCO and LCO are not the crystallographic phases. Correct that, for example, as 'Materials'.

  3. Table 2: d-spacing cannot equal 0. The d-spacing is calculated using Bragg's law: λ=2*dhkl*sin(ÆŸ) for each (hkl) Miller planes. If it is difficult for the authors to calculate d-spacing, the d column should be deleted from table 2.

  4. Line 236: 'they have similar resistance of 19.3 Ω and 19.9 Ω. A.' How were these values obtained? In table 3, the calculated values of ohmic resistance are presented. Use these values for comparison. Provide a link to table 3 in text.

  5. Line 289: correct the name of fig.S3.

Author Response

Dear Editors:

On behalf of my co-authors, we appreciate the editor and reviewers very much for their comments and suggestions on our manuscript entitled “Li7La3Zr2O12-co-LiNbO3 Surface Modification improves Interface Stability between Cathode and Sulfide Solid-State Electrolyte in All-Solid-State Batteries”.

With the rapid development of energy storage and electric vehicles, thiophosphate-based ASSBs are considered the most promising power source. In order to commercialize ASSBs, the interfacial problem between high-voltage cathode active materials and thiophosphate-based solid-state electrolytes needs to be solved in a simple, effective way. Surface coatings are considered the most promising approach to solving the interfacial problem because surface coatings could prevent direct physical contact between cathode active materials and thiophosphate-based solid-state electrolytes. In this work, LLZO and LNO coatings for LCO is fabricated by in-situ interfacial growth of two high Li+ conductive oxide electrolytes on the LCO surface and tested for thiophosphate-based ASSBs. The coatings are obtained from a two-step traditional sol-gel coatings process, the inner coatings are LNO, and the surface coatings are LLZO. Electrochemical evaluations confirm that the two-layer coatings are beneficial for ASSBs. ASSBs containing LLZO&LNO@LCO significantly improve long-term cycling performance and discharge capacity compared to those assembled from uncoated LCO. LLZO&LNO@LCO||LPSC||Li-In delivers discharge capacities of 138.8 mAh/g, 101.8 mAh/g, 60.2 mAh/g, 40.2 mAh/g at 0.05 C, 0.1 C, 0.2 C, 0.5 C under room temperature respectively, and better capacity retentions of 98% after 300 cycles at 0.05 C. The results highlight promising low-cost and scalable cathode materials coatings for ASSBs.

We have studied the reviewers’ comments carefully and have revised. We have tried our best to revise our manuscript according to the comments. Attached please find the revised version, which we would like to resubmit for your kind consideration.

We would like to express our great appreciation to you and reviewers for your comments on our paper. Looking forward to hearing from you.

Thank you and best regards.

Yours sincerely,

Shusen Kang

E-mail: kshusen@163.com

Responds to the reviewer’s comments:

Reviewer: 3

The revised version of the manuscript can be accepted after minor corrections below.

Thanks for your comments.

  1. Methods description was moved to 2 Materials and Methods, but the same text is still in the Results and Discussion. It should be deleted from the Results and Discussion.

Response:

Methods description in Results and Discussion was deleted.

  1. Table 2: the first column is entitled 'phase'. It is incorrect. LLZO&LNO@LCO and LCO are not the crystallographic phases. Correct that, for example, as 'Materials'.

Response:

I correct this question.

  1. Table 2: d-spacing cannot equal 0. The d-spacing is calculated using Bragg's law: λ=2*dhkl*sin(ÆŸ) for each (hkl) Miller planes. If it is difficult for the authors to calculate d-spacing, the d column should be deleted from table 2.

Response:

It is really difficult for me, so I delete the d column.

  1. Line 236: 'they have similar resistance of 19.3 Ω and 19.9 Ω. A.' How were these values obtained? In table 3, the calculated values of ohmic resistance are presented. Use these values for comparison. Provide a link to table 3 in text.

Response:

I correct this problem. Bulk resistance19.3 Ω and 19.9 Ω was record from the EIS curve. Bulk resistance 13.63 Ω and 17.08 Ω was obtained by fitting Nyquist plot.

  1. Line 289: correct the name of fig.S3.

Response:

I correct this question.
